# ChatTracker: Enhancing Visual Tracking Performance via Chatting with Multimodal Large Language Model

**Yiming Sun**[*1,2], **Fan Yu**[*1], **Shaoxiang Chen**[3], **Yu Zhang**[1], **Junwei Huang**[1],
**Yang Li** [1,4] [†], **Chenhui Li**[1], **Changbo Wang** [1]

[1]School of Computer Science and Technology, East China Normal University, Shanghai, China
[2]School of Computer Science, Fudan University, Shanghai, China
[3]Meituan Inc
[4]Shanghai Frontiers Science Center of Molecule Intelligent Syntheses, Shanghai, China.

## Abstract

Visual object tracking aims to locate a targeted object in a video sequence based on an initial bounding box. Recently, Vision-Language (VL) trackers have proposed to utilize additional natural language descriptions to enhance versatility in various applications. However, VL trackers are still inferior to State-of-The-Art (SoTA) visual trackers in terms of tracking performance. We found that this inferiority primarily results from their heavy reliance on manual textual annotations, which include the frequent provision of ambiguous language descriptions. In this paper, we propose ChatTracker to leverage the wealth of world knowledge in the Multimodal Large Language Model (MLLM) to generate high-quality language descriptions and enhance tracking performance. To this end, we propose a novel reflection-based prompt optimization module to iteratively refine the ambiguous and inaccurate descriptions of the target with tracking feedback. To further utilize semantic information produced by MLLM, a simple yet effective VL tracking framework is proposed and can be easily integrated as a plug-and-play module to boost the performance of both VL and visual trackers. Experimental results show that our proposed ChatTracker achieves a performance comparable to existing methods.

## 1 Introduction

Visual object tracking stands as a foundational and challenging task in the computer vision realm [27, 3]. It aims to locate an object in each frame of a video given an initial object box. Recently, Vision-Language (VL) trackers leverage additional natural language descriptions to boost their efficacy. For instance, the shape of a target may change during tracking. However, the semantic information of the target, such as its category or material, remains the same. This makes language text more potential and stable to describe such an appearance-changing object than an image template solely. Despite these advantages, current VL trackers [15, 46, 13] are still inferior to SoTA Visual Trackers [31, 8] on mainstream benchmarks [10, 24]. We identify the following reasons for this: 1) VL trackers heavily rely on manual annotations, which often contain ambiguous language descriptions. 2) Manual textual annotations primarily focus on the tracking target and neglect the semantic information embedded in the text, such as the presence of various background objects and their relations to

---

[*]Equal contribution.
[†]Corresponding author. Email: yli@cs.ecnu.edu.cn

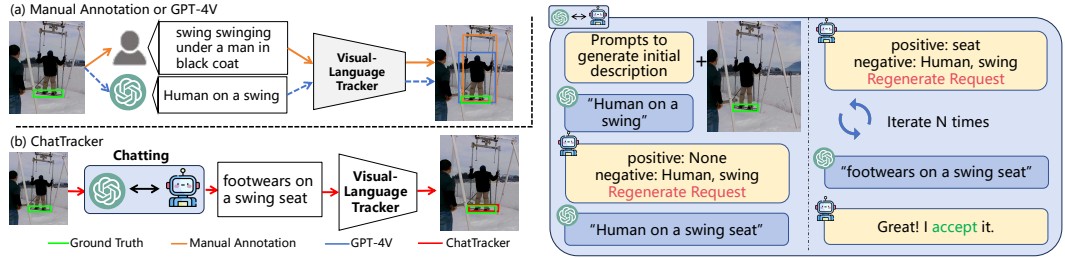

Figure 1: Comparison of different text generation methods. (a) shows manual descriptions and GPT-4V generated descriptions of the tracking target, which are both sub-optimal for tracking. (b) illustrates the generation method used in ChatTracker.

the target. However, most VL trackers mainly focus on better aligning the vision-language modal features [20, 46, 15], overlooking how inaccurate textual annotations in the dataset can adversely affect VL trackers performance. In the last few years, Large Language Models (LLMs) [1] and Multimodal Large Language Models (MLLMs) [29, 48, 38] have progressed rapidly. The wealth of world knowledge encoded in the pre-trained LLMs and MLLMs, along with their capabilities in processing and understanding VL information, has attracted immediate attention from the research communities. Inspired by these advancements, we contemplate whether they can be utilized to achieve better language descriptions for visual tracking. However, we find that directly using the language descriptions generated by MLLMs hardly improves tracking performance as shown in Fig. 1. Two primary causes are identified: 1) The VL tracker is unable to comprehend the language descriptions directly from the MLLM, resulting in the VL trackers identifying incorrect targets. This is because the MLLM and VL trackers are trained on different datasets, leading to a mismatch between the text generated by the MLLM and the visual content in the VL tracker's latent space. 2) The inherent limitations of MLLMs in understanding alternate modalities exacerbate the phenomenon of "hallucination" in multi-modal contexts [7], leading to outputs that are inaccurate or even erroneous.

To address the aforementioned issues, we propose a novel framework, ChatTracker, to integrate MLLMs into visual object tracking. By utilizing the capabilities of MLLMs, a Reflection-based Prompt Optimization (RPO) module is introduced to generate accurate language descriptions of both foreground and background objects. The core idea is to provide feedback to the MLLM about inaccuracies or incomprehensible content of initial language outputs with the VL tracker. This feedback mechanism drives the iterative refinement of the MLLM's output, making it more aligned with the image content and more understandable for the VL tracker, thus effectively addressing the above mentioned issues. In addition, a novel semantic tracking module is proposed to effectively utilize the semantic information obtained from the MLLM and yield the final tracking results. Comprehensive experiments on several widely recognized public datasets are conducted, including LaSOT [10], TrackingNet [24], TNL2K [30], and OTB [16], to demonstrate the effectiveness and efficiency of our proposed method.

Our main contributions are summarized as follows:

1. We propose ChatTracker, a novel framework that leverages MLLMs for visual object tracking. It offers a plug-and-play module enhancement for existing visual and VL trackers with limited computational overhead.

2. We introduce a Reflection-based Prompt Optimization (RPO) module to narrow the knowledge gap between the VL tracker and the MLLM. By reflecting on the feedbacks from tracking, the RPO module can iteratively optimize the prompt for the MLLM and finally produces accurate and relevant descriptions for tracking targets. These descriptions are superior in tracking performance compared to manually annotated texts in datasets.

3. Our proposed ChatTracker achieves comparable performance on several tracking datasets. We conduct extensive experiments including ablation studies to demonstrate the effectiveness of the proposed method and its individual modules.

## 2 Related Work

### 2.1 Vision-Language Trackers

Vision-Language tracking methods [46, 19, 15, 41, 30] have explored the use of linguistic cues to enhance visual object tracking. These approaches can be categorized based on their text sources: those using manually annotated texts and those generating descriptions from a predefined dictionary. In the first category, manually annotated texts have been prevalently employed in target tracking tasks. Datasets like LaSoT [10], TNL2K [30] and MGIT [14] datasets provide manual annotated language descriptions for each sequence. Trackers like the SNLT tracker [11] utilize both visual and language descriptions to predict the target state, then dynamically combine these predictions to produce the final results. JointNLT [46] combines visual grounding and tracking guided by natural language, efficiently addressing the distinct requirements of both processes. The second category leverages a predefined dictionary to generate language descriptions. CiteTracker [15] meticulously develops a category vocabulary that includes attributes like color, texture, and material of the target. During tracking, it uses CLIP [26] to compare the similarity between images and text, selecting the text that closely matches the image as the target's description. In contrast to these approaches, our work exclusively employs MLLM to acquire precise text descriptions of targets. This approach effectively eliminates the reliance on manual text annotations or predefined dictionaries.

### 2.2 Large Language Model in Vision Tasks

Large Language Models (LLMs) like ChatGPT [1] and Llama [29] are auto-regressive models trained on extensive internet-scale text. They encapsulate a vast range of world knowledge within their weights. To integrate visual information into LLMs, various approaches have been developed [48, 6, 40]. Recently, GPT-4V(ision) was released, attracting immediate attention from the community for its outstanding multimodal perception and reasoning capabilities. Its superiority and generality are highlighted in [38]. This has paved the way for a broader spectrum of vision-centric tasks to be addressed. For instance, recent image classification approaches [25, 34, 23, 2], first leverage a LLM to transform class names into more descriptive captions. Following this, the CLIP model is used to classify the images, enhancing the precision of classification tasks. These advancements are primarily directed towards fundamental visual recognition, such as classification and detection. In this work, we are dedicated to integrating the rich world knowledge contained in LLMs into the field of visual object tracking.

## 3 Method

### 3.1 Preliminaries

**Problem Definition.** Given a video $\mathcal{V}$ consisting of $N$ frames: $\{I^t\}_{t=1}^N$, where $I^t$ represents the $t$-th frame of the video. A visual object tracker is tasked to predict bounding boxes $P_{VT}^t$ that tightly wrap the target in further incoming frames with an initial bounding box $G$ (i.e., the position of the target object in the first frame), $P_{VT}^t = \mathcal{F}_{VT}\left(I^t; I^1, G\right)$.

**Multimodal Large Language Models.** A Multimodal Large Language Model (MLLM) $\mathcal{F}_{MLLM}$ takes an image $I \in \mathbb{R}^{H \times W \times 3}$ and a text prompt $T^p$ as inputs, and generates a sequence of textual outputs $T^o$. It can be formulated as: $T^o = \mathcal{F}_{MLLM}\left(I, T^p\right)$. In this paper, we utilize GPT-4V[38], Gemini-1.0 [28] and LLaVA-7B [17] as the MLLM.

**Grounded Visual Language Models.** A Grounded Visual Language Model (GVLM) $\mathcal{F}_{GVLM}$ is designed to accept an image $I \in \mathbb{R}^{H \times W \times 3}$ and a text $T$ with $M$ tokens as inputs, and generates grounded region proposals $P$ and alignment scores $S$ for each token in the text:

$$P, S = \mathcal{F}_{GVLM}(I, T). \tag{1}$$

$P \in \mathbb{R}^{N \times 4}$ denotes the bounding box coordinates of the regions, and $S \in \mathbb{R}^{N \times M}$ is the alignment score, which quantifies the confidence of the model in aligning each word with the corresponding region in the image. $M$ is the number of tokens in the input text and $N$ represents the number of region proposals.

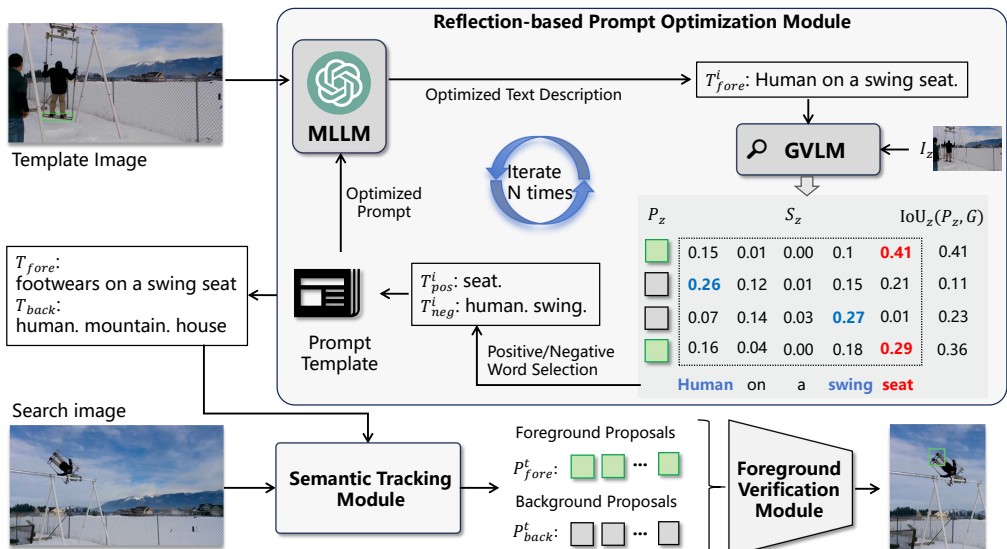

Figure 2: Overall framework of the proposed algorithm. It primarily consists of three parts: A Reflection-based Prompt Optimization Module designed to generate descriptions of both the foreground and background elements to track accurately, a Semantic Tracking Module tasked with creating region proposals for these areas based on the generated descriptions, and a Foreground Verification Module that utilizes these region proposals to select the most precise tracking results. Note that the values in the figure are for visualization and may not match the actual implementation exactly.

## 3.2 ChatTracker Framework

The proposed ChatTracker consists of three components: a Reflection-based Prompt Optimization (RPO) module, a Semantic Tracking module, and a Foreground Verification module. The RPO module takes the template image as input and generates text descriptions of the foreground $T_{fore}$ and background $T_{back}$. Then for each frame $I^t$, the Semantic Tracking module $\mathcal{F}_{ST}$ takes the textual descriptions of both the foreground $T_{fore}$ and background $T_{back}$ as inputs, utilizes a GVLM to obtain foreground region proposals $P^t_{fore}$ and background region proposals $P^t_{back}$:

$$P^t_{fore}, P^t_{back} = \mathcal{F}_{ST}\left(I^t, T_{fore}, T_{back}\right). \qquad (2)$$

The Semantic Tracking module also includes an off-the-shelf single-object visual tracker. We feed it the first frame of the video $I^1$ marked with initial bounding box $G$ and search area $I^t$ and obtain visual tracking results: $P^t_{VT} = \mathcal{F}_{VT}\left(I^t; I^1, G\right)$ for each frame. $P^t_{VT}$ is incorporated as the supplemental foreground proposals into $P^t_{fore}$. Finally, the Foreground Verification module selects the foreground proposal with the highest confidence as the tracking result by considering the their relation with foreground proposals, background proposals, and the template. In the following subsections, we will introduce the details of each module.

## 3.3 Reflection-based Prompt Optimization Module

**Initialization.** We draw a green bounding box on the tracking target in the first frame $I^1$, creating a new image input $I^m$. A pre-defined human-provided prompt template $T_{init}$ along with $I^m$ are input into the MLLM, resulting in initial descriptions of both the foreground and background:

$$T^0_{fore}, T^0_{back} = \text{Extract}\left(\mathcal{F}_{MLLM}\left(I^m, T_{init}\right)\right). \qquad (3)$$

where Extract() refers to the function that reads $T^0_{fore}$ and $T^0_{back}$ from the output text of the MLLM according to a predefined output format. However, due to the hallucination issue of current MLLMs, $T^0_{fore}$ may contain ambiguous language descriptions. Inspired by the successes of LLM reflection [37, 22, 12], we propose a reflection-based iterative method to refine the textual descriptions of the foreground target in case LLMs/MLLMs fail to generate ideal responses in a single attempt.

**Reflection-based Prompt Optimization.** At iteration $i$, the MLLM generates the foreground descriptions $T^i_{fore}$ with $M^i$ words. We input $T^i_{fore}$ and the template image $I^z$ into the GVLM to

obtain grounding results:

$$P_z, S_z = \mathcal{F}_{GVLM}(I^z, T^i_{fore}), \tag{4}$$

where $P_z \in \mathbb{R}^{N \times 4}$ denotes the grounded regions of each target in the template image, and $S_z \in \mathbb{R}^{N \times M}$ is the alignment score between each pair of word and region in the image. Subsequently, based on $P_z$ and $S_z$, we categorize the words $\{w^i_1, w^i_2, ..., w^i_{M^i}\}$ in $T^i_{fore}$ into positive words $T^i_{pos}$ and negative words $T^i_{neg}$:

$$T^i_{pos} = \{w^i_m \mid \exists\, n \text{ such that } S^{nm}_z > \theta_2$$
$$\wedge \text{IoU}(P^n_z,\ G) > \theta_1\}, \tag{5}$$
$$T^i_{neg} = \{w^i_m \mid \text{for all } n \text{ that } S^{nm}_z > \theta_2$$
$$\wedge \text{IoU}(P^n_z,\ G) < \theta_3\} \setminus T^i_{pos}, \tag{6}$$

where $G$ is the ground truth box for target in the template image, and $\text{IoU}(\cdot)$ computes the Intersection-over-Union for two boxes. We define a positive word to be the word that has at least one semantically matching proposal ($S^{nm}_z > \theta_2$) that overlaps significantly with the target ($\text{IoU} > \theta_1$). If all matching proposals of the word can not reach the IoU threshold($\theta_3$), then the word is classified as a negative word. We assess the overall quality of the current set of foreground descriptions $T^i_{fore}$ using the maximum IoU between all proposals and the ground truth $G$, which is denoted as $R^i$. A high value of $R^i$ means that the foreground descriptions can be well-understood by the GVLM to produce accurate grounding results. We set a threshold of $\epsilon$ for $R^i$, and if $R^i > \epsilon$, we use $T^i_{fore}$ as the final foreground description. Otherwise, it indicates that the current $T^i_{fore}$ is inadequate for the GVLM to locate the target. In this case, we construct a new prompt based on the current positive and negative words for the MLLM to generate refined foreground text description:

$$T^{i+1}_{fore} = \mathcal{F}_{MLLM}\left(I^m, \text{Update}(T^i_{pos}, T^i_{neg})\right), \tag{7}$$

where Update indicates filling $T^i_{pos}$ and $T^i_{neg}$ into a pre-defined prompt template provided by humans, resulting in a reflection prompt. Subsequently, this reflection prompt is fed into the MLLM to derive $T^{i+1}_{fore}$. Note that the background description is kept the same since initialization, i.e., $T^{i+1}_{back} = T^i_{back}$. This is because the tracking task lacks background groundtruth, preventing $T^i_{back}$'s iterative optimization. Although it may contain vague language description, $T^i_{back}$ still provides strong semantic information in the tracking scenario. We iterate the above process until the foreground description generated by the MLLM is sufficient for the GVLM to locate the target, i.e., $R^i > \epsilon$, or a maximum number of iterations is reached.

### 3.4 Semantic Tracking Module

After obtaining accurate descriptions, we derive two novel semantic insights previously absent: 1) the relationship between the target and background in the scene, and 2) language descriptions of both foreground and background objects. To utilize these semantic information, we design the Semantic Tracking Module benefiting from MLLMs and the RPO module. Initially, we input the image $I^m$ along with a pre-defined human-provided prompt template into the MLLM. The MLLM then determines whether the target is suitable for tracking using textual information about the relationship between the target and other objects in the scene. If the MLLM deems it unsuitable using textual description, we directly use the visual tracker's prediction $P^t_{VT}$ as the $P^t_{fore}$ and set $P^t_{back}$ to empty. Otherwise we use language descriptions of the foreground $T_{fore}$ and background $T_{back}$ to obtain foreground proposals $P^t_{fore}$ and background proposals $P^t_{back}$. We first perform grounding using the concatenated words of $T_{fore}$ and $T_{back}$ on the template image $I^z$:

$$P_z, S_z = \mathcal{F}_{GVLM}(I^z, \mathcal{C}(T_{fore}, T_{back})), \tag{8}$$

where $\mathcal{C}(\cdot)$ denotes the word concatenation operation with each word separated by '.'. Then tokens associated with bounding boxes that exhibit a high IoU score with the ground truth $G$ (exceeding threshold $\theta_1$) are classified as foreground tokens $V_{fore}$. Conversely, tokens linked to bounding boxes with a low IoU score (below threshold $\theta_3$) are categorized as background tokens $V_{back}$: $V_{fore} = \{v_m \mid \exists\, n \text{ such that } S^{nm}_z > \theta_2 \wedge \text{IoU}(P^n,\ G) > \theta_1\}, V_{back} =$

$\{v_m \mid \exists\, n \text{ such that } S_z^{nm} > \theta_2 \wedge \text{IoU}(P^n,\ G) < \theta_3\} \setminus V_{fore}$. Note that $v_m$ is the m-th token (one word may contain multiple tokens) in $\mathcal{C}(T_{fore}, T_{back})$. We categorize foreground and background by tokens instead of words because we empirically found it leads to better performance in semantic grounding and tracking. After the foreground and background tokens are divided, they are fixed and used during the tracking of all subsequent frames. For the $t$-th frame, we obtain region proposals: $P^t, S^t = \mathcal{F}_{GVLM}(I^t, \mathcal{C}(T_{fore}, T_{back}))$. Then, using $V_{fore}$ and $V_{back}$, we classify the region proposals $P^t$ into foreground proposals $P_{fore}^t$ and background proposals $P_{back}^t$: $P_{fore}^t = \{P_n^t \mid \exists\, m \text{ such that } S_{mn}^t > \theta_2 \wedge v_m \in V_{fore}\}$, $P_{back}^t = \{P_n^t \mid \exists\, m \text{ such that } S_{mn}^t > \theta_2 \wedge v_m \in V_{back}\}$. Because $P_{fore}^t$ may be empty, we additionally incorporate the result of the visual tracker $P_{VT}^t$ as the supplemental foreground proposals into $P_{fore}^t$.

### 3.5 Foreground Verification Module

To further select result in $P_{fore}^t$, we compute two types of metrics: $W_{fore}$ and $W_{back}$. The $W_{fore}$ is determined based on the similarity between the proposal and the template, whereas $W_{back}$ assesses the proposal's relationship with the background proposals. The final score is then established through a combination of $W_{fore}$ and $W_{back}$.

**Foreground Scorer.** Motivated by [44], we trained a neural network $f(\cdot)$ with generated foreground and background proposals to map the target template and foreground proposals into a discriminative Euclidean space. Its loss function is as follows: $\sum_{i=1}^{K}\left[\|f(\mathcal{X}_i^a) - f(\mathcal{X}_i^p)\|_2^2 - \|f(\mathcal{X}_i^a) - f(\mathcal{X}_i^n)\|_2^2 + \alpha\right]_+$, where $\mathcal{X}_i^a$ denotes the $i$-th bounding box of a specific target, $\mathcal{X}_i^p$ a positive sample of identical target in other frames, and $\mathcal{X}_i^n$ is a negative sample of any other target or background. $\alpha$ is a margin value. During inference, we can determine the foreground scores $W_{fore} = \{s_{fore}^1, s_{fore}^2, ..., s_{fore}^N\}$ of the foreground proposals $P_{fore}^t = \{P_{fore}^{t1}, P_{fore}^{t2}, ..., P_{fore}^{tN}\}$ using a cosine similarity metric as $s_{fore}^i = \max\left(\text{similarity}\left(f(z), f\left(\phi\left(P_{fore}^{ti}\right)\right)\right), 0\right)$, where $z$ represents the target template in the first frame and $\phi\left(P_{fore}^{ti}\right)$ is the image cropped within the bounding box $P_{fore}^{ti}$.

**Background Scorer.** To enhance tracking performance by incorporating background information, we develop a background scorer. This scorer scores a foreground proposal $P_{fore}^t$ by its maximum IoU with all the background proposals $P_{back}^t$. During the inference stage, background scores $W_{back} = \{s_{back}^1, s_{back}^2, ..., s_{back}^N\}$ are computed as follows:

$$s_{back}^i = \max_j\left(\text{IoU}\left(P_{fore}^{ti}, P_{back}^{tj}\right)\right). \tag{9}$$

Finally, the overall score $W_{all} = \left[s^1, s^2, \ldots, s^N\right]$ is determined by combining the foreground and background scores:

$$s^i = s_{fore}^i \times (1 - s_{back}^i). \tag{10}$$

In the final step, the foreground proposal $\mathbf{b}_i$ with the highest score $s^i$ is selected as the output of the tracking process.

## 4 Experiment

### 4.1 Experimental Settings

Our experiments are conducted an NVIDIA 3090 GPUs. The alignment score threshold $\theta_2$ is set to 0.2, while the IoU thresholds for foreground and background, $\theta_1$ and $\theta_3$, are set to 0.3 and 0.1, respectively. In the proposed RPO module, $\epsilon$ is set to 0.4. We adopt GPT4V-preview1106 [38] as our default MLLM, GroundingDINO-T [18] as the GVLM. We use MixFormer and ARTrack as the visual trackers for ChatTracker-L and ChatTracker-B, respectively. ChatTracker-L is designed for better performance, while ChatTracker-B is designed to achieve a better trade-off between accuracy and speed. We used UVLTrack-B [20] in place of STM and FVM in ChatTracker-B.

Table 1: **State-of-the-art comparisons on the datasets of TNL2K, LaSOT and TrackingNet.**
The best two results are shown in red and blue color. Our approach performs favorably against
the state-of-the-art methods on all datasets. $^*$ indicates vision-language trackers. All metrics of
performance are in % in tables unless otherwise specified.

| Method | Source | LaSOT | | | TrackingNet | | | TNL2K | | |
|---|---|---|---|---|---|---|---|---|---|---|
| | | AUC | $P_{Norm}$ | P | AUC | $P_{Norm}$ | P | AUC | $P_{Norm}$ | P |
| ChatTracker-L | Ours | 74.1 | 83.8 | 81.2 | 86.1 | 90.3 | 86.0 | 65.4 | 76.5 | 70.2 |
| ChatTracker-B | Ours | 71.7 | 80.9 | 77.5 | 83.6 | 88.1 | 82.2 | 59.6 | 76.3 | 62.1 |
| UVLTrack-B$^*$ [21] | AAAI2024 | 69.4 | - | 74.9 | 83.4 | - | 82.1 | 63.1 | 80.9 | 66.7 |
| CiteTracker$^*$ [15] | ICCV2023 | 69.7 | 78.6 | 75.7 | 84.5 | 89.0 | 84.2 | 57.7 | 73.6 | 59.6 |
| DecoupleTNL$^*$ [19] | ICCV2023 | 71.2 | - | 75.3 | - | - | - | 56.7 | - | 56.0 |
| JointNLT$^*$ [46] | CVPR2023 | 60.4 | 69.4 | 63.6 | - | - | - | 56.9 | 73.5 | 58.1 |
| RGFM-B256 [47] | NeurIPS2023 | 70.3 | 82.0 | 76.4 | 84.7 | 89.6 | 83.6 | - | - | - |
| MixformerV2-B [9] | NeurIPS2023 | 70.6 | 80.8 | 76.2 | 83.4 | 88.1 | 81.6 | 57.4 | - | 58.4 |
| F-BDMTrack-384 [36] | ICCV2023 | 72.0 | 81.5 | 77.7 | 84.5 | 89.0 | 84.0 | 57.8 | - | 59.4 |
| MITS [33] | ICCV2023 | 72.0 | 80.1 | 78.5 | 83.4 | 88.9 | 84.6 | - | - | - |
| ARTrack-384 [31] | CVPR2023 | 72.6 | 81.7 | 79.1 | 85.1 | 89.1 | 84.8 | 59.8 | - | - |
| SeqTrack-L384 [4] | CVPR2023 | 72.5 | 81.5 | 79.3 | 85.5 | 89.8 | 85.8 | 57.8 | - | - |
| DropTrack [32] | CVPR2023 | 71.8 | 81.8 | 78.1 | 84.1 | 88.9 | - | 56.9 | - | 57.9 |
| MATTracker [43] | CVPR2023 | 67.8 | 77.3 | - | 81.9 | 86.8 | - | 51.3 | - | - |
| MMTrack$^*$ [45] | TCSVT2023 | 70.0 | 82.3 | 75.7 | - | - | - | 58.6 | 75.2 | 59.4 |

## 4.2 Comparison with Existing Trackers

As shown in Table 1, we compare ChatTracker against 5 state-of-the-art vision-language trackers and
8 state-of-the-art visual trackers on three popular datasets [10, 24, 30].

**LaSOT** [10] is a large-scale, long-term single object tracking benchmark with 280 videos, each
averaging more than 2,448 frames. And each video includes a phrase simply describing the tracking
target. On this dataset, ChatTracker-L achieves the top-tier performance, with an AUC of 74.1%,
surpassing JointNLT by a large margin of **13.5%**. This again proves the text generated by iterative
refinement of the MLLM can enhance the tracker's understanding of both the target and the overall
scene, which leads to improvements on long term tracking performance.

**TrackingNet** [24], a prominent large-scale benchmark for short-term object tracking, comprises an
extensive collection of 30,643 video segments. But it does not include textual annotations describing
the target. In this challenging dataset, ChatTracker-L has achieved a remarkable AUC of 86.1%,
surpassing all previous trackers. This performance demonstrates our tracker's superior capability in
handling diverse and dynamic short-term tracking scenarios. It it noteworthy that, the lack of textual
annotations in the TrackingNet test set renders most vision-language trackers ineffective. However,
our ChatTracker demonstrates its unique ability to adapt to this scenario, thus broadening the scope
of vision-language tracking applications.

**TNL2K** [30] is a benchmark designed for evaluating vision-language tracking algorithms. And each
video here contains a more detailed phrase describing the tracking target. Compared to the recent
vision-language tracker JointNLT, which utilizes both the annotated sentences and bounding box, our
approach surpasses it by **12.1%** in precision. This indicates that with the proposed RPO module, our
tracker is able to utilize optimized textual descriptions for more accurate tracking.

## 4.3 Generalization and Universality

Our proposed framework can act as a plug-and-play solution that boosts the performance of both
visual trackers and VL trackers, demonstrating superior generalization capabilities.

**For visual trackers**, we have integrated four distinct visual trackers with ChatTracker-B. Table 2
shows that this integration results in significant performance improvements for all trackers. This
suggests that our proposed framework can be generally used with other existing visual trackers to
boost their performance.

**For VL trackers**, we replaced the manually annotated text inputs from the dataset with the foreground
descriptions generated by ChatTracker. These results in Table 3 demonstrate that ChatTracker can
generate language descriptions that are more accurate than manually annotated descriptions and can
effectively boost tracking performance in general.

Table 2: Results of Visual Trackers with the integration of ChatTracker (marked by [+]). All results are measured on the same device.

| Methods | LaSOT | | | TNL2K | | | OTB-lang | | |
|---|---|---|---|---|---|---|---|---|---|
| | AUC | P | $P_{Norm}$ | AUC | P | $P_{Norm}$ | AUC | P | $P_{Norm}$ |
| OStrack-256 [39] | 69.11 | 75.22 | 78.68 | 54.16 | 53.12 | 69.02 | 69.20 | 90.39 | 83.64 |
| **OStrack-256[+]** | **70.23** | **76.49** | **80.04** | **56.51** | **56.85** | **72.06** | **69.69** | **90.72** | **84.20** |
| TransT-N4 [5] | 64.85 | 69.02 | 73.78 | 53.16 | 54.26 | 69.70 | 69.55 | 90.61 | 84.25 |
| **TransT-N4[+]** | **67.34** | **72.38** | **76.73** | **56.27** | **58.47** | **73.09** | **70.06** | **90.63** | **84.51** |
| Stark-S [35] | 65.78 | 69.73 | 75.15 | 53.10 | 51.95 | 68.90 | 67.25 | 86.92 | 81.70 |
| **Stark-S[+]** | **66.76** | **71.29** | **76.35** | **55.63** | **56.12** | **71.59** | **67.93** | **87.81** | **82.60** |

Table 3: The comparison of results for Vision-Language trackers using ChatTracker-generated text (marked by *). We report the AUC value on the datasets.

| Methods | LaSOT | OTB-lang | TNL2K |
|---|---|---|---|
| JointNLT [46] | 56.74 | 58.57 | 54.38 |
| **JointNLT*** | **57.96** | **60.07** | **54.82** |
| UVLTrack [21] | 56.55 | 59.39 | 54.78 |
| **UVLTrack*** | **57.23** | **59.98** | **55.61** |

Table 4: Text-to-image alignment scores for manually annotated and ChatTracker-generated language descriptions. ViT and RN refer to the use of ViT-B/32 and RN-50 as CLIP [26] image encoders, respectively.

| Source of text | LaSOT | TNL2K | OTB-lang |
|---|---|---|---|
| Manual-ViT | 24.74 | 23.58 | 23.13 |
| **ChatTracker-ViT** | **24.87** | **23.93** | **23.67** |
| Manual-RN | 18.03 | 17.57 | 16.87 |
| **ChatTracker-RN** | **18.46** | **18.13** | **17.41** |

**With different MLLM**. We also replace GPT-4V with gemini-1.0-pro-vision-latest [28] and LLaVA-7B [17] in the ChatTracker-B to study whether our method can adapt to different MLLMs (both proprietary and open-source). As shown in Table 5, these MLLMs generally lead to performance improvements, and surprisingly, the results of adopting LLaVA-7B are comparable with proprietary MLLMs. This shows the effectiveness of our method itself regardless of the choice of the MLLM.

### 4.4 Analysis on Language Descriptions Generated by ChatTracker.

To further validate the generation of high-quality language descriptions of our proposed method, we conduct image-text matching experiments. Specifically, we cropped the target from each frame and calculated its text-to-image alignment scores [26] with both manually annotated textual descriptions and ChatTracker-generated descriptions. In Table 4, we report the maximum text-to-image alignment score during the iteration process for each sequence. As shown in Table 4, ChatTracker-generated descriptions have better text-to-image correlation compared with manually annotated descriptions across three datasets. Such advancements highlight the potential for enhancing future vision-language trackers by providing more accurate language descriptions.

### 4.5 Ablation Study

To validate the effectiveness of the proposed modules, we perform ablation studies on three variants of our model. **Base Model** exclusively employs the visual tracker to perform the tracking task. In the ablation study, we use TransT-N4 [5] as the visual tracker. **w/o RPO** utilizes manually annotated text from the dataset as input for semantic tracking. Due to the absence of background descriptions generated by the RPO module, we only generate foreground proposals $P_{fore}^t$, and use $W_{fore}$ to select the tracking results. **w/o ITER** solely utilizes the foreground and background text descriptions generated from the first iteration of the RPO module for tracking.

First, w/o RPO achieves similar performances with Base Model on all three datasets, which indicates that manually annotated text is sub-optimal for performing vision-language tracking. Then, comparing w/o ITER with Base Model or w/o RPO, we observe a modest enhancement across three datasets. The improvements validate the effectiveness of our proposed Semantic Tracking module and Foreground Verification module. The textual descriptions directly obtained from an MLLM might be inaccurate

Table 5: Results of ChatTracker-B using various MLLMs. BaseTracker is ARTracker-256 [31].

| Methods | LaSOT | | TNL2K | | OTB-lang | |
|---|---|---|---|---|---|---|
| | AUC | $P_{Norm}$ | AUC | $P_{Norm}$ | AUC | $P_{Norm}$ |
| BaseTracker | 70.77 | 79.54 | 58.09 | 74.33 | 69.90 | 84.10 |
| GPT-4V | 71.68 | 80.92 | 59.63 | 76.27 | 70.77 | 85.29 |
| Gemini1.0 | 70.98 | 80.13 | 60.23 | 76.97 | 70.96 | 85.61 |
| LLaVA-7B | 71.36 | 80.54 | 59.90 | 76.49 | 70.86 | 85.57 |

Table 6: Ablation study of the proposed algorithm. The best results in each part of the table are marked in **bold**.

| | | Base Model | w/o RPO | w/o ITER | Ours |
|---|---|---|---|---|---|
| LaSOT | AUC | 64.85 | 64.63 | 64.87 | **67.89** |
| | P | 69.02 | 68.80 | 69.12 | **73.07** |
| | $P_{Norm}$ | 73.78 | 73.75 | 73.95 | **77.18** |
| TNL2K | AUC | 53.16 | 55.70 | 53.66 | **56.39** |
| | P | 54.26 | 57.27 | 54.65 | **58.76** |
| | $P_{Norm}$ | 69.70 | 72.60 | 70.28 | **73.03** |

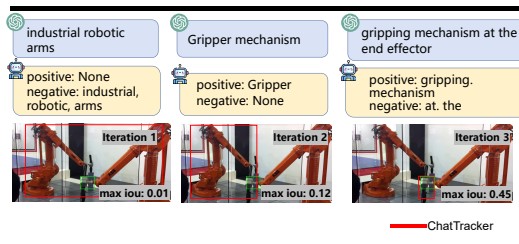
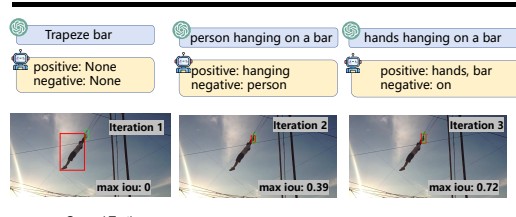

Figure 3: Illustrations of prompt optimization in a dialogue scenario. Each set shows the initial manual annotation, the subsequent prompts generated by the LLM, and the final optimized prompt that successfully guided the Vision-Language tracker to the target.

and noisy. Nevertheless, our approach effectively mitigates this noise by utilizing the semantic matching ability of the GVLM. Furthermore, when we compare our method with Base Model and w/o ITER, there is a notable performance improvement across all datasets. These improvements demonstrate that our proposed RPO module can generate accurate descriptions of the target by utilizing the rich knowledge of the MLLM, and the generated descriptions are even better than the manually annotated ones provided by the datasets. Finally, the performance gain of our method over w/o ITER verifies the effectiveness of iterative optimization of the RPO module.

### 4.6 Qualitative Study

To better understand the effectiveness of the proposed RPO module, we illustrate the process of prompt optimization in a dialogue scenario in Fig. 3.

The example on the right is from the *Swing-14* sequence, where a person is depicted mid-air, gripping a horizontal bar, which is the tracking target. However, the dataset's manual annotation describes it as "swing swinging above a man in black pants". Without context, even humans might struggle to identify the target using this description. Initially, the MLLM, drawing from its extensive knowledge, accurately describes the object as a "Trapeze bar". Yet, due to the knowledge gap between the vision-language tracker and the MLLM, the tracker fails to locate the target. After receiving the feedback, the MLLM rephrases its output with simple terms like "person" and "bar". It then uses the semantic context of "hanging" to assist the tracker in target identification. At this stage, the tracker can approximately locate the target. However, since the IoU is below the preset threshold of 0.4, it sends "hanging" back as a positive sample and "person" as a negative sample to the MLLM for further refinement. In the final iteration, the MLLM pinpoints "hands" as the critical term. This term is both easy to understand and consistently visible on the target throughout the sequence.

### 4.7 Limitations

When the tracking target is in low resolution or lacks discernible visual features, the MLLM struggles to provide an accurate language description of the target. Additionally, accessing the MLLM via API necessitates an internet connection, which may pose challenges in edge deployments due to intermittent or unreliable network access.

Temporal changes to target and background are one of the challenges in the visual tracking domain. However, we do not update language descriptions for two key reasons. First, the tracker's predictions are not always accurate, and there are no annotations for background objects, making it harder to generate prompts dynamically. Second, calling MLLMs multiple times during tracking to update these prompts adds much computational cost. Achieving a good balance between performance and efficiency requires extensive research.

Additionally, our ChatTracker focuses solely on the visual features of the tracking target, such as shape, texture, and color, without addressing the impact of different granularities of text annotations as discussed in MGIT [14].

## 5 Conclusion

In this work, we introduced ChatTracker, the first method that utilizes the Multimodal Large Language Model (MLLM) to enhance the performance of visual tracking. We proposed a Reflection-based Prompt Optimization (RPO) module to iteratively refine the ambiguous and inaccurate language descriptions of the target with tracking feedback. Moreover, a simple yet effective visual-language tracking framework was proposed to boost the performance of existing trackers as a plug-and-play method. Experimental results on multiple datasets demonstrated that our method outperformed state-of-the-art methods. This suggests that incorporating MLLMs into visual tracking had a notable effect on improving tracking performance.

**Acknowledgements** This work was supported by the National Natural Science Foundation of China (62102152, 62072183), and the Shanghai Urban Digital Transformation Special Fund Project (202301027). This work was also sponsored by the Shanghai Frontiers Science Center of Molecule Intelligent Syntheses.

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

# Appendix

## A  Broader Impact

In this paper, we introduced ChatTracker, an efficient and precise tracking framework that integrates MLLM. The remarkable efficiency and effectiveness of ChatTracker enable its seamless integration into monitoring systems for unauthorized observations. The proposed PRO Module, through the LLM feedback mechanism, bridges the knowledge gap between the vision language trackers and the MLLM, allowing the MLLM to better adapt to the downstream vision language tracking task. We believe this offers valuable insights for addressing the knowledge gap between visual and textual modalities.

## B  Analysis on Language Descriptions Generated by ChatTracker.

In this section, we present the specific methods for calculating image-text alignment. For a tracking dataset $V$ that contains $N$ video sequences: $V = \{v_1, v_2, \ldots, v_N\}$, each video sequence $v_k$ consists of $m_k$ frames: $v_k = \{I_0^k, I_1^k, \ldots, I_{m_k}^k\}$. In the $k$-th sequence, $I_j^k$ refers to the $j$-th frame, and the corresponding tracking result is represented by $Y_j^k$. Each sequence is accompanied by a manually annotated textual description $T_k^a$ and a foreground description generated by ChatTracker, $T_k^c$. Based on these tracking results, we extract image regions from $I_j^k$ to form: $C_k = \{C_0^k, C_1^k, \ldots, C_{m_k}^k\}$. The text-to-image alignment score for the manually annotated descriptions ($S_a$) and the ChatTracker-generated descriptions ($S_c$) are computed as follows:

$$S_a = \frac{1}{N} \sum_{k=1}^{N} \frac{1}{m_k} \sum_{j=1}^{m_k} \frac{f_i(C_j^k) \cdot f_t(T_k^a)}{\|f_i(C_j^k)\|_2 \cdot \|f_t(T_k^a)\|_2}, \tag{11}$$

$$S_c = \frac{1}{N} \sum_{k=1}^{N} \frac{1}{m_k} \sum_{j=1}^{m_k} \frac{f_i(C_j^k) \cdot f_t(T_k^c)}{\|f_i(C_j^k)\|_2 \cdot \|f_t(T_k^c)\|_2}. \tag{12}$$

Here, $f_t$ and $f_i$ represent CLIP's [26] text and image feature extractors, respectively. $S_a$ and $S_c$ indicate the degree of alignment between textual and the tracking target. Notably, ChatTracker-generated descriptions consistently outperform manually annotated descriptions across three datasets. Such advancements highlight the potential for enhancing future visual-language trackers by providing more accurate language descriptions.

When background information is present in the language description, the image-text similarity is lower compared to language descriptions without background information. Therefore, the higher the image-text similarity between the language descriptions and the cropped target, the less background information is included in the language descriptions. The less background information included in the language descriptions indicates that the text quality is higher. This aligns with the observation that higher alignment scores ($S_c$) demonstrate the improved capability of ChatTracker in generating more target-focused descriptions, further reducing background interference.

## C  Visualized Results

Figure 4 shows the visualized results of CiteTracker [15] and Our ChatTracker on two datasets with a total of six sequences. It is clear to see that the accuracy of our method greatly outperforms that of CiteTracker.

## D  Supplimentary Experiments

### D.1  More about Foreground Verification module

The Foreground Verification module is crucial to the whole framework and its final performance.To better prove this, we design an additional ablation experiment using ChatTracker-L on the OTB-Lang dataset. The experiment involves three versions of our trackers: **Complete ChatTracker-L** uses the

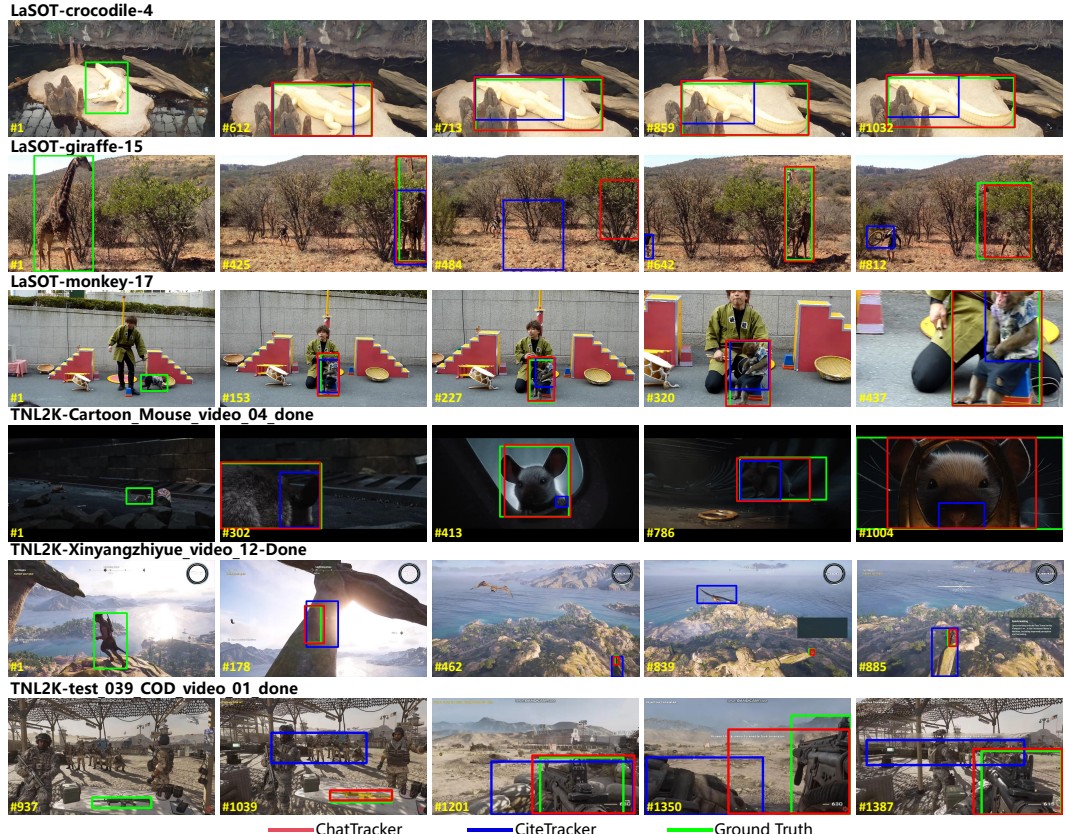

Figure 4: Visualized results of the proposed algorithm and the CiteTracker method on six challenging sequences with drastic changes. ChatTracker demonstrates superior performance. In contrast, CiteTracker faces difficulties in handling these complex sequences.

Table 7: The performance comparison of Foreground Verification module ,random selection module in our framework on OTB-lang Dataset

| Method | AUC | P | $\mathbf{P}_{Norm}$ |
|---|---|---|---|
| ChatTracker-L | 71.78 | 94.25 | 86.82 |
| ChatTracke-L_random_sample | 42.98 | 58.74 | 52.11 |
| ChatTracke-L_upperbound | 73.91 | 96.17 | 88.61 |

Foreground Verification module.**ChatTracker-L with Ground Truth** does not use the Foreground Verification module but selects the proposal with the highest IoU with Ground Truth as the tracking result. This entry is used to establish the theoretical upper bound.**ChatTracker-L with Random Selection** does not use the Foreground Verification module and randomly selects a proposal as the tracking result. The results are shown in the Table 7. The results show that using the Foreground Verification module significantly outperforms the random selection group. This demonstrates that our Foreground Verification module is effective. Although there may still be room for improvement, our ChatTracker-L result is close to the theoretical upper bound.

## D.2 More performance comparison

We conduct performance comparison on OTB-Lang dataset.

Table 8: Results on the OTB-Lang dataset

| Method | AUC | P | $P_{Norm}$ |
|---|---|---|---|
| ChatTracker-L | 71.78 | 94.25 | 86.82 |
| ChatTracker-B | 70.77 | 92.00 | 85.29 |
| JointNLT | 65.52 | 86.23 | 80.41 |
| ARTrack-256 | 69.90 | 91.15 | 84.10 |

And we also conduct more SoTA comparison with other trackers.

Table 9: More state-of-the-art comparisons on the datasets of TNL2K and LaSOT. And $*$ indicates vision-language trackers.

| Method | Source | LaSOT | | | TNL2K | | |
|---|---|---|---|---|---|---|---|
| | | AUC | $P_{Norm}$ | P | AUC | $P_{Norm}$ | P |
| ChatTracker-L | Ours | 74.1 | 83.8 | 81.2 | 65.4 | 76.5 | 70.2 |
| ChatTracker-B | Ours | 71.7 | 77.5 | 80.9 | 59.6 | 76.3 | 62.1 |
| $VLT^*_{TT}$ [13] | NeurIPS2022 | 67.3 | 77.6 | 72.1 | 53.1 | - | 53.3 |
| OVLM-384* [42] | TMM2023 | 67.7 | 77.6 | 74.2 | 64.7 | 82.6 | 69.3 |
| OVLM-256* [42] | TMM2023 | 65.6 | 75.6 | 71.1 | 62.5 | 80.0 | 66.5 |

