# OpenReview forum: "ChatTracker: Enhancing Visual Tracking Performance via Chatting with Multimodal Large Language Model"
_NeurIPS.cc/2024/Conference — NeurIPS 2024 poster_

### Official Review · Reviewer_Bwy6 · 2024-06-21

**Soundness:** 3
**Presentation:** 3
**Contribution:** 3
**Rating:** 5
**Confidence:** 4

**Summary:**

This paper introduces a simple yet effective VL tracking framework based on a multimodal large language model, called ChatTracker. The main idea is to utilize the rich world knowledge in multimodal large language model to generate high quality language descriptions and improve tracking performance. Specifically, a reflection-based prompt optimization module is proposed to iteratively refine the ambiguous and inaccurate descriptions of the target with tracking feedback. A large number of experiments demonstrate the effectiveness of the proposed method.

**Strengths:**

1.the paper is well written and can be easily understood by the reader.

2.The proposed tracker can achieve SOTA performance and adequate ablation experiments are accomplished.

**Weaknesses:**

- Line 127 mentions only the initial bounding box and search frame as inputs and does not mention template frames as inputs. However, the equation in line 128 writes template as input.

- In section 4.1, MixFormer-L and ARTrack-256 as visual coders for ChatTracker-L and ChatTracker-B? I have a question, why not use mixformer-B/L or ARTrack-B/L uniformly as visual encoders. Could the authors please provide more results for mixformer-B and ARTrack-L for performance comparison.

- In section 4.3 and Table 2, when applying the proposed module to other visual trackers, the paper does not describe how the linguistic description is fused with the visual features, could the authors please explain this process specifically.

- The LaSOT_ext and OTB_Lang test datasets are also included in the VL tracking task, and the authors are asked to provide performance on these datasets to help enrich the experimental results in this paper.

**Questions:**

- It is an interesting idea to apply large language models to the visual tracking field. The general practice would be to do fine-tuning operations on the downstream data so that the large language model is more adapted to the downstream task. I have a question, does the proposed approach require fine-tuning in the downstream tracking dataset?

**Limitations:**

The limitations of the method have been accounted for in this paper.

---

> ### Author Rebuttal · Authors · 2024-08-04
>
> ***Q1: Line 127 mentions only the initial bounding box and search frame as inputs and does not mention template frames as inputs. However, the equation in line 128 writes template as input.***
>
> Thanks for your careful reading.
>
> In the equation  $P^{t}_{VT} = \mathcal{F}_V$$_T$$(I^{t};I^{1}, G) $ ,  $I^t$  represents the image of the t-th frame, $G$ represents the initial bounding box, and $I^1$ is the first frame of the video with  $G$ as the initial bounding box.
> We will clarify this in the revised manuscript.
>
> ***Q2: In section 4.1, MixFormer-L and ARTrack-256 as visual coders for ChatTracker-L and ChatTracker-B? I have a question, why not use mixformer-B/L or ARTrack-B/L uniformly as visual encoders. Could the authors please provide more results for mixformer-B and ARTrack-L for performance comparison.***
>
> Thanks for your question.
>
> At the time of submission, ARTrack[1] had not released the code and model weights for ARTrack-L384, so we did not use ARTrack-L as the visual encoder for ChatTracker-L.The choice of MixFormer-L[2] for ChatTracker-L leads to high accuracy, and ARTrack-B for ChatTracker-B provides a better trade-off between accuracy and speed.
>
> We provide more results for MixFormer-B and ARTrack-L for performance comparison. In the table below, results of visual trackers with the integration of ChatTracker are marked by *.
>
> **Table.1  The comparison of results forVision Language trackers using ChatTracker-generated text  (marked by\*). The speed was tested on the same device.**
>
> |                           | LaSOT[3] |       |       | TNL2K[4] |       |       | Speed (fps) |
> | ------------------------- | ----------- | ----- | ----- | ------------ | ----- | ----- | ----------- |
> | Choice for visual tracker | $AUC$ | $P$  | $P_{Norm}$    | $AUC$  | $P$     | $P_{Norm}$    |             |
> | Mixformer-B               | 69.15        | 74.70 | 78.70 | 55.12        | 55.17 | 71.21 | 57          |
> | Mixformer-B*              | 70.43        | 76.27 | 80.28 | 57.38        | 57.93 | 73.94 | 44          |
> | Mixformer-L               | 70.1         | 76.3  | 79.9  | 60.55        | 63.01 | 76.16 | 27          |
> | Mixformer-L*              | 74.1         | 83.8  | 81.2  | 65.41        | 70.20 | 83.25 | 20          |
> | ARTrack-256               | 70.77        | 76.23 | 79.54 | 58.09        | 59.90 | 74.33 | 51          |
> | ARTrack-256*              | 71.68        | 77.50 | 80.92 | 59.63        | 62.06 | 76.27 | 48          |
> | ARTrack-L384              | 73.49        | 80.56 | 82.34 | 60.58        | 64.36 | 77.25 | 22          |
> | ARTrack-L384*             | 74.65        | 82.07 | 84.71 | 62.01        | 65.65 | 78.93 | 18          |
>
>
> ***Q3: In section 4.3 and Table 2, when applying the proposed module to other visual trackers, the paper does not describe how the linguistic description is fused with the visual features, could the authors please explain this process specifically.***
>
> In fact, we do not fuse the visual tracker into our ChatTracker at the feature level. When replacing visual trackers, we incorporate the different tracking results of the corresponding visual tracker, $P^t_{VT}$, into $P^t_{fore}$ as supplemental foreground proposals. At the same time, the foreground proposals generated by the GVLM using linguistic descriptions do not change.
>
> We will provide a detailed description of how we replace the other visual tracking methods in the revised manuscript and we will release our code to help our readers better understand this process.
>
>
> ***Q4: The LaSOT_ext and OTB_Lang test datasets are also included in the VL tracking task, and the authors are asked to provide performance on these datasets to help enrich the experimental results in this paper.***
>
> Thank you for your suggestion. Our performance on these two datasets is as follows, demonstrating that our method also achieves SOTA results on these datasets.
>
> **Table.2  Performance comparison of ChatTracker, UVLTrack, and ARTrack on the LaSOT_ext and OTB_Lang datasets.**
>
> |               | LaSOT_ext |       |       | OTB_Lang[5] |       |       |
> | ------------- | --------- | ----- | ----- | --------------- | ----- | ----- |
> |               | $AUC$       | $P$     | $P_{Norm}$    | $AUC$             | $P$     | $P_{Norm}$    |
> | ChatTracker-L | 56.06     | 64.84 | 68.18 | 71.78           | 94.25 | 86.82 |
> | ChatTracker-B | 53.72     | 62.09 | 65.13 | 70.77           | 92.00 | 85.29 |
> | UVLTrack-L    | 51.21     | 59.00 | 62.30 | 71.89           | 93.21 | 87.77 |
> | ARTrack-L384  | 52.8      | 59.7  | 62.9  | 71.66           | 92.80 | 85.78 |
> | ARTrack-256   | 48.36     | 53.73 | 57.69 | 69.90           | 91.15 | 84.10 |
>
> ***Q5: Does the proposed approach require fine-tuning in the downstream tracking dataset?***
>
> No, we don't need to fine-tune MLLM in the downstream tracking dataset. We use the MLLM reflection to narrow the knowledge gap between the VL tracker and the MLLM. This allows ChatTracker to achieve favorable performance without fine-tuning.
>
> Thanks for your suggestion. We will investigate the fine-tuning version of our proposed framework in future work.
>
>
> [1]Xing Wei, et al.  Autoregressive visual  tracking. In Proceedings of the IEEE/CVF Conference on Computer Vision and Pattern  Recognition,2023.
>
> [2] Yutao Cui, et al.  Mixformer: End-to-end tracking with  iterative mixed attention. In Proceedings of the IEEE/CVF Conference on Computer Vision and  Pattern Recognition,2022.
>
> [3]Heng Fan, et al.  Lasot: A high-quality large-scale single object tracking benchmark.  International Journal of Computer Vision, 2021.
>
> [4]Xiao Wang, et al.  Towards more flexible and accurate object tracking with natural language: Algorithms  and benchmark. In Proceedings of the IEEE/CVF Conference on Computer Vision and Pattern  Recognition, 2021.
>
> [5]Zhenyang Li, et al.   Tracking by natural language specification. In Proceedings of the IEEE conference on computer  vision and pattern recognition, 2017

---

> > ### Comment · Reviewer_Bwy6 · 2024-08-12
> > **Comment**
> >
> > Thanks to the author's reply, my concerns were largely addressed. I will maintain my original rating score.

---

> ### Author Response · Authors · 2024-08-12
> **Thanks to Reviewer Bwy6**
>
> Dear Reviewer,
>
> Thanks a lot for your recognition of our work.
>
> Best wishes,
>
> The Authors

---

### Official Review · Reviewer_z5a5 · 2024-07-12

**Soundness:** 2
**Presentation:** 3
**Contribution:** 2
**Rating:** 5
**Confidence:** 4

**Summary:**

This paper proposes ChatTracker, a novel framework that leverages MLLMs for visual object tracking. The Reflection-based Prompt Optimization (RPO) module can narrow the knowledge gap between the VL tracker and the MLLM. ChatTracker also achieves SoTA performance on several tracking datasets.

**Strengths:**

1. This paper designs a new framework that leverages MLLMs for visual object tracking. The entire framework's process is clear and easy to follow.
2. The approach performs favorably against the state-of-the-art methods on several datasets.

**Weaknesses:**

1. SOTA trackers are missing in Table 1, such as OVLM (TMM23), MMTrack(TCSVT23), etc. There is a discrepancy between the data from UVLTrack in Table 1 and the data provided by the official source.
2. I want to know the model's performance on MGIT (NeurIPS '23) and OTB99_Lang (CVPR '17) to verify the generalizability of the method in different scenarios (short-term tracking and global instance tracking).
3. Why is it necessary to design ChatTracker-L and ChatTracker-B, but use different backbone networks? Also, I believe comparing the L model in Table 1 is unfair, as UVLTrack provides the L model, and annotations are needed for models of different scales.
4. For analysis of language descriptions generated by ChatTracker, if the natural language description includes background information, is it reasonable to only crop out the target for calculating image-text similarity?
5. I believe the phrase "via Chatting with Multimodal Large Language Model" in the title overstates the contribution of the entire framework.

**Questions:**

Please refer to weaknesses.

**Limitations:**

Yes

---

> ### Author Rebuttal · Authors · 2024-08-04
>
> ***Q1: SOTA trackers are missing in Table 1, such as OVLM[1], MMTrack[2].***
>
> Thank you for your valuable suggestion. We will include comparisons with OVLM and MMTrack in the revised manuscript. Additionally, we have found that compared to their variants with highest performance, our ChatTracker demonstrates better tracking performance. Furthermore, our method does not require manually annotated target text for initialization.
>
> **Table.1** Complete comparison of OVLM and MMTrack with ChatTracker's performance on LaSOT[3] and TNL2K[4] Datasets.
>
> | Method| LaSOT | | TNL2K | |
> |-|-|-|-|-|
> | | $AUC$ | $P$    | $AUC$   | $P$ |
> | MMTrack | 70.0  | 75.7 | 58.6  | 59.4 |
> | OVLM-256 | 65.6  | 71.1 | 62.5  | 66.5 |
> | OVLM-384 | 67.7  | 74.2 | 64.7  | 69.3 |
> | ChatTracker-B | 71.7  | 80.9 | 59.6  | 62.1 |
> | ChatTracker-L | 74.1  | 81.2 | 65.4  | 70.2 |
>
> ***Q2:  I want to know the model's performance on MGIT[5] and OTB99_Lang[6].***
>
> Thank you for your valuable suggestion. We evaluated ChatTracker-B on MGIT and OTB99_Lang.
>
> - **For the MGIT dataset**, we used only the first frame's bounding box (BBOX) for initialization and used ChatTracker's self-generated language descriptions for tracking. For the comparison method JointNLT[7], we selected Mechanism A (i.e., the first frame's BBOX and the annotated text in the dataset) for comparison. For ARTrack-256[8], we used only the first frame's BBOX for initialization.
>
> **Table.2**. Results on the MGIT dataset
>
> | Tracker | **Normalized Precision (N-PRE)** | **Precision (PRE)** | S**uccess Rate (SR_IoU)**. |
> |-|-|-|-|
> | ChatTracker-B | 0.833 | 0.691  | 0.801 |
> | JointNLT      | 0.786 | 0.445  | 0.610 |
> | ARTrack-256   | 0.711 | 0.516   | 0.599 |
>
> - **For the OTB99_Lang dataset,** our ChatTracker used only the first frame's bounding box (BBOX) for initialization and utilized ChatTracker's self-generated language descriptions for tracking.  The settings for JointNLT and ARTrack-256 are the same as those for MGIT.
>
> **Table.3**: Results on the OTB99_Lang dataset
> | Tracker    | $AUC$   | $P$     | $P_{Norm}$    |
> |-|-|-|-|
> | ChatTracker-B | 70.77 | 92.00 | 85.29 |
> | JointNLT      | 65.52 | 86.23 | 80.41 |
> | ARTrack-256   | 69.90 | 91.15 | 84.10 |
>
> The results show that, thanks to more accurate text descriptions and the proposed vision-language tracking framework, our ChatTracker outperforms existing methods. We will include these results in the revised manuscript.
>
> ***Q3 Why is it necessary to design ChatTracker-L and ChatTracker-B, but use different backbone networks?***
>
> - For the design of ChatTracker-L and ChatTracker-B:
>
> ChatTracker-L is designed for better performance, while ChatTracker-B is designed to achieve a better trade-off between accuracy and speed. Here, L stands for Large, and B stands for Base.
>
> - For the  use of different backbone networks:
>
> At the time of submission, ARTrack[1] had not released the training code and model weights for ARTrack-L384, so we use the Mixformer-L for ChatTracker-L.
>
> ***Q4: There is a discrepancy between the data from UVLTrack[9] in Table 1 and the data provided by the official source. Also, I believe comparing the L model in Table 1 is unfair, as UVLTrack provides the L model, and annotations are needed for models of different scales.***
>
> Thank you for pointing this out.  In Table 1, we reported the results for UVLTrack-B. As shown in the table below, our AUC scores on LaSOT, TrackingNet, and TNL2K are higher than UVLTrack-L. For other trackers in Table 1, we selected the variant with the highest performance for a fair comparison. We will replace the comparison results of UVLTrack-B in Table 1 with the UVLTrack-L version and annotate the scales of the compared methods in the revised manuscript. We have included the revised Table.1 in the uploaded PDF file.
>
> **Table.4 Comparison results of UVLTrack-L and ChatTracker on the LaSOT, TrackingNet, and TNL2K datasets**
>
> | | LaSOT |      |  | TrackingNet | |  | TNL2K |      |
> |-|-|-|-|-|-|-|-|-|
> |   | $AUC$ | $P$  |  | $AUC$ | $P$  |  | $AUC$ | $P$  |
> | UVLTrack-L    | 71.3  | 78.3 |  | 84.1  | 82.9 |  | 64.8  | 68.8 |
> | ChatTracker-L | 74.1  | 81.2 |  | 86.1 | 86.0 |  | 65.4  | 70.2 |
>
>
> ***Q5:  if the natural language description includes background information, is it reasonable to only crop out the target for calculating image-text similarity?***
>
> It is reasonable to only crop out the target for calculating image-text similarity.
>
> When background information is present in the language description, the image-text similarity is lower compared to language descriptions without background information. Therefore, the higher the image-text similarity between the language descriptions and the cropped target, the less background information is included in the language descriptions. The less background information included in the language descriptions indicates that the text quality is higher.
>
> Therefore, in our experiments, cropping the target from each frame and calculating its text-to-image similarity is reasonable. We will clarify this in the revised manuscript.
>
> ***Q6:  I believe the phrase "via Chatting with Multimodal Large Language Model" in the title overstates the contribution of the entire framework.***
>
>  "via Chatting with Multimodal Large Language Model"  refers to the iterative process where the MLLM generates language descriptions, and the GVLM provides feedback on the quality of these descriptions to the MLLM, helping it generate better language descriptions. This process is akin to chatting with the MLLM. We think this phrasing helps our readers better understand our method intuitively.
>
> We are open to changing the title if there is a more suitable one. We would appreciate it if you could give more detailed suggestion or more explanation about how does the current one overstate the contribution.
>
> Due to the word limit in the rebuttal, we provide the remaining citations in the following comment.

---

> ### Author Response · Authors · 2024-08-04
> **Response to Reviewer z5a5（part2）**
>
> Due to the word limit in the rebuttal, we provide the remaining citations here.
>
> [1]OVLM: Huanlong Zhang, Jingchao Wang, Jianwei Zhang, Tianzhu Zhang, Bineng Zhong "One-stream Vision-Language Memory Network for Object Tracking" TMM 2023
>
> [2]MMTrack: Yaozong Zheng, Bineng Zhong, Qihua Liang, Guorong Li, Rongrong Ji, Xianxian Li"Towards Unified Token Learning for Vision-Language Tracking" TCSVT 2023
>
> [3]Heng Fan, Hexin Bai, Liting Lin, Fan Yang, Peng Chu, Ge Deng, Sijia Yu, Harshit, Mingzhen  Huang, Juehuan Liu, et al. Lasot: A high-quality large-scale single object tracking benchmark.  International Journal of Computer Vision, 129:439–461, 2021.
>
> [4]Xiao Wang, Xiujun Shu, Zhipeng Zhang, Bo Jiang, Yaowei Wang, Yonghong Tian, and Feng  Wu. Towards more flexible and accurate object tracking with natural language: Algorithms  and benchmark. In Proceedings of the IEEE/CVF Conference on Computer Vision and Pattern  Recognition, pages 13763–13773, 2021.
>
> [5]Shiyu Hu, Dailing Zhang, Meiqi Wu, Xiaokun Feng, Xuchen Li, Xin Zhao, and Kaiqi Huang. 2023. "A Multi-modal Global Instance Tracking Benchmark (MGIT): Better Locating Target in Complex Spatio-temporal and Causal Relationship." Paper presented at the 37th Conference on Neural Information Processing Systems (NeurIPS 2023), Track on Datasets and Benchmarks.
>
> [6]Y. Wu, J. Lim and M. -H. Yang, "Object Tracking Benchmark," in IEEE Transactions on Pattern Analysis and Machine Intelligence, vol. 37, no. 9, pp. 1834-1848, 1 Sept. 2015, doi: 10.1109/TPAMI.2014.2388226.
>
> [7] Li Zhou, Zikun Zhou, Kaige Mao, and Zhenyu He. Joint visual grounding and tracking with  natural language specification. In 2023 IEEE/CVF Conference on Computer Vision and Pattern  Recognition (CVPR), pages 23151–23160, 2023.
>
> [8]Xing Wei, Yifan Bai, Yongchao Zheng, Dahu Shi, and Yihong Gong. Autoregressive visual  tracking. In Proceedings of the IEEE/CVF Conference on Computer Vision and Pattern  Recognition, pages 9697–9706, 2023.
>
> [9]Yinchao Ma, Yuyang Tang, Wenfei Yang, Tianzhu Zhang, Jinpeng Zhang, and Mengxue Kang.  Unifying visual and vision-language tracking via contrastive learning, 2024

---

> > ### Comment · Reviewer_z5a5 · 2024-08-10
> > **Official Comment by Reviewer z5a5**
> >
> > I read the author's experiments and replies to all reviewers. I am grateful for the author's efforts during the rebuttal, which largely answered my doubts. There are still some small details that need to be paid attention to. On the one hand, MGIT provides flexible and diverse semantic labels. This benchmark is somewhat closer to the "chat" feature than current VLT benchmarks. Therefore, I suggest that the author conduct further experiments on this benchmark to analyze the effects and limitations of the method under different granularities and evaluation mechanisms, and consider adding this part of the analysis to the article to highlight the motivation and theme of the article. In addition, please note that the author cited the wrong MGIT literature (Reference 5) in the second reply window for me. I am unsure what the relationship is between the work cited by the author and the rebuttal itself. Finally, NeurIPS officially recommends that the reply to each reviewer during the rebuttal should not exceed 6,000 characters (only one window). I noticed that the author's reply to 2 reviewers had exceeded the limit of one window. In a sense, this is unfair (for other authors of the same period who only use one window for rebuttal). I hope the authors can pay attention to these details. In summary, I would like to temporarily maintain my original score and listen to the opinions of other reviewers on the authors' rebuttal.

---

> ### Author Response · Authors · 2024-08-10
> **Thanks and Response to Review z5a5**
>
> Dear Reviewer:
>
> Thank you again for taking the time to review our work and for providing us with valuable feedback. We now address your additional comments.
>
> ***Q1.  MGIT provides flexible and diverse semantic labels. This benchmark is somewhat closer to the "chat" feature than current VLT benchmarks. Therefore, I suggest that the author conduct further experiments on this benchmark to analyze the effects and limitations of the method under different granularities and evaluation mechanisms, and consider adding this part of the analysis to the article to highlight the motivation and theme of the article.***
>
> We appreciate your valuable suggestion. MGIT is a high-quality vision-language benchmark. The MGIT dataset focuses on the impact of text annotations at different granularities on tracking performance.
>
> However, our ChatTracker only uses the initial bounding box as input and does not utilize additional text descriptions as input.
> Although we agree that further applying our proposed method to MGIT is very interesting, employing different text annotation granularities is beyond the scope of this paper.
>
> In the rebuttal, we have provided ChatTracker's results on the MGIT dataset with the initial bounding box and we will include these results in the revised manuscript.
>
> Thank you again for your valuable suggestion. We will further investigate these text inputs into our framework in future work.
>
> ***Q.2 In addition, please note that the author cited the wrong MGIT literature (Reference 5) in the second reply window for me. I am unsure what the relationship is between the work cited by the author and the rebuttal itself.***
>
> Thank you for your careful reading. We have corrected Reference [5] in the comment.
>
> ***Q.3 Finally, NeurIPS officially recommends that the reply to each reviewer during the rebuttal should not exceed 6,000 characters (only one window). I noticed that the author's reply to 2 reviewers had exceeded the limit of one window. In a sense, this is unfair (for other authors of the same period who only use one window for rebuttal). I hope the authors can pay attention to these details.***
>
> Thanks again for your advice. For additional comments, we follow the [NeurIPS 2024] Clarification on author rebuttal email:
>
> > - Comments to paper and reviews will be fine. Comments can be seen in time. Please set the readers correctly when you post them. Reviewers are not required to take comments into consideration.
>
> To better discuss the issues, we responded to the longest review (up to 11 questions) in one additional Comments and provided relevant references in another Comment.
>
> We strictly follow the policy and instructions given by the NeurIPS 2024 PCs email. This email is sent to all authors and we don't think there is any fairness issue.
>
> We hope our responses have fully addressed your concerns. If you have any other questions or need further clarification, please feel free to let us know.
>
> Thank you very much!
>
> Best wishes,
>
> Authors

---

### Official Review · Reviewer_aCWh · 2024-07-13

**Soundness:** 3
**Presentation:** 3
**Contribution:** 3
**Rating:** 6
**Confidence:** 2

**Summary:**

The paper proposes a novel Multimodal Large Language Model framework to improve the vision-language visual tracking performance.
By introducing the reflection-based prompt optimization module, the tracking prompt can be iteratively refined via tracking feedback.
The proposed method shows state-of-the-art results compared to prior vision-language trackers and visual trackers.

**Strengths:**

- The paper investigates the multimodal large language models for the visual tracking problem.
- By showing the limitations of prompts created by MLLM and manual annotation for visual tracking, the paper introduces a novel iterative prompt generation via chatting and semantic prompt verification.
- The paper shows adequate evaluation comparing the proposed method with both vision-language trackers and visual trackers.

**Weaknesses:**

The paper should illustrate the number of chat iterations in the reflection-based prompt optimization module and their performance effect. The example in Figure 5 shows that the module needs 2 iterations to get an accepted prompt but the overall analysis should be considered, especially in the images with complex scenes and objects.

**Questions:**

Please refer to the weakness.

**Limitations:**

The authors have mentioned the limitations.

---

> ### Author Rebuttal · Authors · 2024-08-04
>
> ***Q1:   The paper should illustrate the number of chat iterations in the reflection-based prompt optimization module and their performance effect.  The example in Figure 5 shows that the module needs 2 iterations to get an accepted prompt but the overall analysis should be considered, especially in the images with complex scenes and objects.***
>
> -  The paper should illustrate the number of chat iterations in the reflection-based prompt optimization module.
>
> Thanks for your advice. The iterative refinement process averages **2.3** rounds. We illustrated the number of chat iterations in Section 4.1 (Line 229 - 230).
>
> -  The example in Figure 5 shows that the module needs 2 iterations to get an accepted prompt but the overall analysis should be considered, especially in the images with complex scenes and objects.
>
> We conducted experiments using ChatTracker-B to study the performance effect of the maximum number of iterations. The results show that when the maximum number of iterations is less than 10, the tracker performance improves with an increase in iterations. When the number of iterations exceeds 10, the improvement in tracker performance becomes less significant.  In our experiments, the default maximum number of iterations is set to 20.
>
> For complex scenes and objects, we will add more visual examples in the revised manuscript to help our readers understand the impact of iteration numbers on tracker performance.
>
> **Table.1  Impact of the number of iteration rounds on tracker performance on Lasot[1] and TNL2k[2] datasets**
>
> | Number of Maximum Iterations | 1     | 10    | 20    | 30    |
> | ---------------------------- | ----- | ----- | ----- | ----- |
> | LaSOT AUC                    | 64.87 | 67.24 | 67.89 | 67.91 |
> | TNL2K AUC                    | 53.66 | 55.63 | 56.39 | 56.42 |
>
> [1]Heng Fan, Hexin Bai, Liting Lin, Fan Yang, Peng Chu, Ge Deng, Sijia Yu, Harshit, Mingzhen  Huang, Juehuan Liu, et al. Lasot: A high-quality large-scale single object tracking benchmark.  International Journal of Computer Vision, 129:439–461, 2021.
>
> [2]Xiao Wang, Xiujun Shu, Zhipeng Zhang, Bo Jiang, Yaowei Wang, Yonghong Tian, and Feng  Wu. Towards more flexible and accurate object tracking with natural language: Algorithms  and benchmark. In Proceedings of the IEEE/CVF Conference on Computer Vision and Pattern  Recognition, pages 13763–13773, 2021.

---

> > ### Comment · Reviewer_aCWh · 2024-08-12
> >
> > Thank the authors for their response. I am satisfied with the answer. After carefully reading other reviews, I will maintain the score.

---

> > > ### Author Response · Authors · 2024-08-12
> > > **Thanks to  Reviewer aCWh**
> > >
> > > Dear Reviewer,
> > >
> > > Thank you very much for recognizing our work.
> > >
> > > Best wishes,
> > >
> > > The Authors

---

### Official Review · Reviewer_gPPX · 2024-07-13

**Soundness:** 3
**Presentation:** 3
**Contribution:** 3
**Rating:** 7
**Confidence:** 4

**Summary:**

The paper proposes a new Visual-Language (VL) tracking framework called ChatTracker, that integrates MLLMs into VL tracking through iterative refinement of text prompts for VL trackers. The text prompts optimized using the proposed Reflection-based Prompt Optimization (RPO) module are more accurate than manual text annotations in the datasets and improve the tracking performance by taking into account both foreground and background objects.

**Strengths:**

- The paper identifies a clear gap for Visual-Language trackers and how manually annotated or naively generated text prompts can provide sub-optimal results for language-assisted visual tracking.
- The proposed framework combines the power of a strong MLLM like GPT-4V and a GVLMs like Grounding DINO-T to formulate the iterative refinement procedure for generating better text prompts for tracking tasks. It uses a GVLM powered by the optimized prompts in conjunction with a visual tracker to get a set of proposals. It also incorporates GVLM generated region proposals to train its Foreground Verification module, which not only does foreground classification but scores candidate regions with how they overlap with background proposals. While different parts of the framework are pre-existing and straight-forward, the paper uses them in a new way to improve visual tracking performance in relevant benchmarks.
- The method is plug-and-play, and any visual tracker (regardless of whether it is a VL tracker or not) can be used as the baseline tracker in the Semantic Tracking Module of the framework.
- The experimental results and ablations support the claims in the paper.
- The paper is well-written, well-organized and clear.

**Weaknesses:**

- The proposed framework does not take into account the potential temporal changes in the video that might require a more optimal foreground/background text prompt to proceed. For example, in Fig. 3 we see the framework landing on ‘hands, bar’ as the positive prompt, which focuses on hands being attached to the bar, which may not be true for the rest of the video. The background text of the video is also never updated. In fact, neither the GVLM that the framework heavily depends on nor the Foreground Verification module have any temporal components. Therefore, the framework would have to heavily rely on the baseline tracker in this aspect.
- The quality of the region proposals coming out of the Semantic Tracking Module is un-addressed. How often GVLM proposals are preferred to the visual tracker results (unless the prompt optimization fails as mentioned in Appendix A)? How dependent is the final performance on how well the Foreground Verification module is performing (since it’s only been trained on the LaSOT training set)?

**Questions:**

- The potential limitations mentioned in the weaknesses should be addressed or discussed in the paper. Temporal changes to target and background are the main challenges in visual tracking and the proposed framework does not add any mechanisms to address these even though it might heavily depend on GVLM results for its answers.
- Related to the above point: Have the authors done any experiments where foreground/background prompts were updated while the video is being processed? What were the insights?
- As mentioned above, how often does the visual tracker proposal get picked over a GVLM proposal and vice versa when both exist in the pool going into the Foreground module? Or does the visual tracker proposal mostly get used when there are no GVLM proposals?
- Related works section can be expanded a bit. An important motivation for this paper is the disadvantages brought by the manual textual annotations in the VOT benchmarks (that have them). For the broader audience, the authors could add a short section on related benchmarks and if they have manual text annotations, explain how those came about (why are they low quality?). Line [300-301] mentions the prompt “swing swinging above a man in black pants” - which almost sounds like a generated text prompt, having this context in the paper itself would be helpful.
- Equation (6) - What’s the purpose of $T^i_{pos}$ in this equation?
- Lines [152-153] - What happens if a word matches to no proposals? Can we add that to negative words? Why not?
- Lines [204-205] - To confirm, the anchor sample is always the target template, but do you sample positive samples from tracker result or foreground proposal? Visual tracking is mentioned in Line 202, but not in the Appendix.
- Line [221] - The IoU threshold for especially foreground seems a little low, what’s the reason for this? Do you have any statistics for what the accepted IoUs are from the foreground verification? Is it generally lower with the proposals?
- Please use $P_{norm}$ or NP consistently across tables. Also, for a broader audience you should mention what the metrics are even if it's in a couple sentences.
- Some tables report NP and P, others one or the other. It would be better to report consistently even if it’s just across smaller tables. Moreover, baseline tracker results in Table 3 are not matching the same tracker’s results in Table 1. If I’m missing some details, please let me know.
- Fig. 3 - not seeing any blue boxes in the image, even though CiteTracker is in the legend.

**Limitations:**

Authors did address some limitations of their method in the Appendix. Please see my discussions about other potential limitations in the weaknesses and questions sections above.

---

> ### Author Rebuttal · Authors · 2024-08-04
>
> ***Q1.  The potential limitations mentioned in the weaknesses should be addressed or discussed in the paper.  Have the authors done any experiments where foreground/background prompts were updated while the video is being processed? What were the insights?***
>
> Thanks for pointing this out. Temporal changes to target and background are indeed one of the main challenges in the visual tracking domain. Our ChatTracker relies on a baseline tracker to consider potential temporal changes in the video. However, compared to traditional visual trackers, which rely only on a single image template to locate the target, **natural language provides more temporal-invariant features**, such as object categories and textures. Our ChatTracker incorporates MLLMs and designs a novel Reflection-based Prompt Optimization (RPO) module to obtain accurate target descriptions, providing better robustness when facing temporal changes.
>
> Updating foreground and background prompts is not straightforward for two main reasons:
>
> 1. **Accuracy of Predictions**: During tracking, the tracker's predictions of the target are not always accurate, and there are no annotations of background objects. This makes it difficult to dynamically generate foreground and background prompts.
> 2. **Trade-off Between Performance and Efficiency**: If MLLMs are called multiple times during tracking to update foreground and background prompts, it incurs a certain computational overhead (although this overhead is decreasing as technology advances). Achieving a good balance between performance and efficiency requires extensive research.
>
> We intend to delve into this challenge as part of our future work, and to address this, we have incorporated a discussion in the limitations section.
>
>
>
>
>
> ***Q2.  How often does the visual tracker proposal get picked over a GVLM proposal?  Or does the visual tracker proposal mostly get used when there are no GVLM proposals?***
>
> A  GVLM proposal gets picked 33.27% of the all time in our experiment. We analyzed 685,360 frames in the LaSOT dataset **[1]**, of which 228,042 used the GVLM proposal as the result.
>
>
>
>
>
> ***Q3. How dependent is the final performance on how well the Foreground Verification module is performing (since it’s only been trained on the LaSOT training set)?***
>
> Thanks for pointing this out. The Foreground Verification module is crucial to the whole framework and its final performance. To better answer your question, we design an additional ablation experiment using ChatTracker-L on the OTB-Lang dataset[2]. The experiment involves three versions of our trackers:
>
> 1. **Complete ChatTracker-L**: Uses the Foreground Verification module.
> 2. **ChatTracker-L with Ground Truth**: Does not use the Foreground Verification module but selects the proposal with the highest IoU with Ground Truth as the tracking result. This entry is used to establish the theoretical upper bound.
> 3. **ChatTracker-L with Random Selection**: Does not use the Foreground Verification module and randomly selects a proposal as the tracking result.
>
> The results are shown in the table below. The results show that using the Foreground Verification module significantly outperforms the random selection group. This demonstrates that our Foreground Verification module is effective. Although there may still be room for improvement, our ChatTracker-L result is close to the theoretical upper bound. Thank you for your suggestion. We will discuss and add this in the supplementary.
>
> **Table.1  The performance comparison of Foreground Verification module ,random selection module in our framework on Lasot Dataset**
>
> | Method| $AUC$ | $P$| $P_{Norm}$ |
> |-|-| - |-|
> | ChatTracker-L| 71.78 | 94.25 | 86.82 |
> | ChatTracke-L_random_sample | 42.98 | 58.74 | 52.11 |
> | ChatTracke-L_upperbound | 73.91 | 96.17 | 88.61 |
>
>
>
>
>
> ***Q4 Related works section can be expanded a bit.***
>
> Thanks for your suggestions. We will add a section in the revised manuscript to introduce related benchmarks for broader audiences to understand the insights. This section will include whether the LaSOT, TrackingNet, TNL2K, and other benchmarks contain text annotations, the methods used for text annotation, and potential reasons for low quality.
>
>
> ***Q5 What’s the purpose of  $T_{pos}^i$ in equation  (6) ?***
>
> In equation (6):
>
>  $ T_{neg}^i = \\{  w_m^i|for\ all\ n\ that\ S_z^{nm}>\theta_2 \wedge IoU(P_z^n,G)<\theta_3 \\}  \backslash T_{pos}^i$
>
> The purpose of    $T_{pos}^i$  is to ensure that a word does not appear in both  $T_{pos}^i$  and $T_{neg}^i$ .
>
> $T_{pos}^i$ and $T_{neg}^i$  are used to construct a reflection prompt during the iterative optimization process to help the MLLM generate better language descriptions. If a word is present in both$T_{pos}^i$  and $T_{neg}^i$ , it may confuse the MLLM and prevent it from generating an effective response.
>
> ***Q6 Lines [152-153] - What happens if a word matches to no proposals? Can we add that to negative words? Why not?***
>
> If a word matches no proposals, it will be ignored and not added to negative words.
>
>  $T_{neg}^i$ is used to inform the LLM that the GVLM has associated this word with objects in the background. The LLM will further enable the GVLM to identify the correct target based on this information.
>
> When a word matches no proposals, it is considered meaningless (i.e., it doesn't contain any information that helps the LLM distinguish the target), so we do not add it to  $T_{neg}^i$ .
>
> For example, in the swing-17 example in Fig. 2, when the text description is “Human on a swing seat,” "Human" and "Swing" are categorized as negative words, and "seat" is categorized as positive. They all correspond to image regions in the figure. However, "on" and "a," which match no proposals, do not provide effective feedback to the MLLM. Therefore, we do not add them to negative words.
>
> Due to the word limit in the rebuttal, we provide the remaining responses  in the following comment.

---

> ### Author Response · Authors · 2024-08-04
> **Response to Reviewer  gPPX（part2）**
>
> Due to the word limit in the rebuttal, we provide the remaining responses here.
>
> ***Q7  Lines [204-205] - To confirm, the anchor sample is always the target template, but do you sample positive samples from tracker result or foreground proposal? Visual tracking is mentioned in Line 202, but not in the Appendix.***
>
> No, we do not sample positive samples from the visual tracker's results during training. We only randomly choose a target patch from other frames of the same video as the positive sample. In Line 202, we use the trained $ f(.) $​ (which is a neural network) to calculate the foreground score of the visual tracker's result. We will clarify this in the revised manuscript.
>
>
>
> ***Q8 Line [221] - The IoU threshold for especially foreground seems a little low, what’s the reason for this? Do you have any statistics for what the accepted IoUs are from the foreground verification? Is it generally lower with the proposals?***
>
> -  The IoU threshold for especially foreground seems a little low, what’s the reason for this? Is it generally lower with the proposals?
>
> For the IoU threshold for foreground(i.e. $\theta_1$ in equation (5)  ), we set $\theta_1$  to 0.3. This is to ensure there are enough positive words  $T_{pos}^i$  as feedback during the early iterations, helping the iterative optimization process converge quickly.
>
> For accepted IoUs (i.e., $\epsilon$ in line157) ，we choose the IoU threshold based on the experiments. We tested on the LaSOT dataset, and the results are shown in the table below. The results indicate that an IoU threshold (i.e., $\epsilon$ in line157) of 0.4 leads to the best results.
>
> **Table.2  The performance of chattracker on Lasot dataset with different IoU thresholds**
>
> | IoU threshold | 0.3   | 0.4   | 0.5   | 0.6   |
> | ------------- | ----- | ----- | ----- | ----- |
> | AUC           | 69.87 | 71.68 | 71.54 | 70.92 |
>
> - Do you have any statistics for what the accepted IoUs are from the foreground verification?
>
> Yes, the average accepted IoU is 0.79. We calculated the distribution of accepted IoUs on the LaSOT dataset, and the results are shown in the table below.
>
> **Table.3 Data distribution of IoU values on the Lasot dataset.**
>
> | IoU        | [0,0.4) | [0.4,0.6) | [0.6,0.8) | [0.8, 1.0]|
> | ------------------ | ------- | --------- | --------- | ---------- |
> | number.of.sequence | 20      | 14        | 52        | 194        |
>
> Thank you for your question. We will add the relevant results and analysis in the revised manuscript.
>
>
>
>
>
>
>
> ***Q9  Please use $P_{norm}$ or NP consistently across tables.   Some tables report NP and P, others one or the other.  It would be better to report consistently even if it’s just across smaller tables.***
>
> Thanks for pointing this out. In the revised manuscript, we will use $P_{norm}$ or NP consistently and report metrics consistently.
>
>
>
>
>
> ***Q10  baseline tracker results in Table 3 are not matching the same tracker’s results in Table 1.***
>
> Thanks for pointing this out.
>
> In Table 1, the baseline trackers (i.e., JointNLT[3] and UVLTrack[4] ) are initialized using both natural language and the initial bounding box to compare with the best-performing model.
>
> In Table 3, the baseline trackers (i.e., JointNLT and UVLTrack) are initialized using only natural language. The purpose is to compare the effectiveness of ChatTracker-generated text versus manually annotated text for Vision-Language trackers without the influence of the initial bounding box.
>
> We will clarify this in the revised manuscript.
>
> ***Q11  Fig. 3 - not seeing any blue boxes in the image, even though CiteTracker is in the legend.***
>
> Thanks for pointing out this typo. CiteTracker[5] is not included in this comparision and we reuse the legend picture in this figure mistakenly. We revise the manuscript accordingly. For clarification, we have placed the revised figure in the .pdf file.

---

> ### Author Response · Authors · 2024-08-04
> **Response to Reviewer gPPX（part3）**
>
> Due to the word limits, we provide the remaining citations here.
>
> [1]Heng Fan, Hexin Bai, Liting Lin, Fan Yang, Peng Chu, Ge Deng, Sijia Yu, Harshit, Mingzhen  Huang, Juehuan Liu, et al. Lasot: A high-quality large-scale single object tracking benchmark.  International Journal of Computer Vision, 129:439–461, 2021.
>
> [2]Y. Wu, J. Lim and M. -H. Yang, "Object Tracking Benchmark," in IEEE Transactions on Pattern Analysis and Machine Intelligence, vol. 37, no. 9, pp. 1834-1848, 1 Sept. 2015, doi: 10.1109/TPAMI.2014.2388226.
>
> [3] Li Zhou, Zikun Zhou, Kaige Mao, and Zhenyu He. Joint visual grounding and tracking with  natural language specification. In 2023 IEEE/CVF Conference on Computer Vision and Pattern  Recognition (CVPR), pages 23151–23160, 2023.
>
> [4]Yinchao Ma, Yuyang Tang, Wenfei Yang, Tianzhu Zhang, Jinpeng Zhang, and Mengxue Kang.  Unifying visual and vision-language tracking via contrastive learning, 2024
>
> [5]Xin Li, Yuqing Huang, Zhenyu He, Yaowei Wang, Huchuan Lu, and Ming-Hsuan Yang.  Citetracker: Correlating image and text for visual tracking. In Proceedings of the IEEE/CVF  International Conference on Computer Vision, pages 9974–9983, 2023.

---

> > ### Comment · Reviewer_gPPX · 2024-08-11
> >
> > I want to thank the authors for all their efforts in responding to my and other reviewers' questions and comments. The analyses the authors provided were really helpful for clarifying the mechanics of their method for me. After reading the other reviews and the authors' responses to them, I'm more comfortable with my assessment of the paper. Thanks to the authors for including all this new information in their revised manuscript. I do agree with reviewer z5a5's comment that it would very valuable to analyze the proposed method on MGIT's multi-granular annotations.

---

> ### Author Response · Authors · 2024-08-12
> **Thanks to Reviewer gPPX**
>
> Dear Reviewer,
>
> Thank you once again for your valuable comments. We also agree with reviewer z5a5's advice on this further research direction. It would be very interesting to extend our framework in MGIT's multi-granular annotations and dataset.
>
> Since analyzing the effect of multi-granular annotations is non-trivial and beyond the scope of this paper, we will investigate this direction in our future work.
>
> Thank you very much!
>
> Best wishes,
>
> The Authors

---

### Author Rebuttal · Authors · 2024-08-05

Dear All Reviewers,

Thank you again for taking the time to review our work and for providing us with valuable feedback.

We are excited that you found our results impressive (gPPX, z5a5, Bwy6) and our experiments well-designed (gPPX, aCWh, Bwy6), appreciate the innovative uses of multimodal large language models for visual tracking (gPPX, aCWh, z5a5), and found our paper well-written (gPPX, Bwy6). We will open-source the code to facilitate a better understanding of our method. The revised Table 1 and Figure 3 are included in the PDF submitted with this response.

We have carefully considered your comments and have provided our responses. If you have any further questions or require additional clarification, please kindly let us know.

Thank you again for your valuable input.

Best wishes,
Authors

---

### Decision · Program_Chairs · 2024-09-25

**Decision:**

Accept (poster)

**Comment:**

The rebuttal provided clarifications about the proposed method and its analysis that were useful for assessing the paper's contribution and responded adequately to most reviewer concerns. All reviewers recommend acceptance after discussion (with two borderline accepts, one weak accept and one accept), and the ACs concur. The final version should include all reviewer comments, suggestions, and additional clarifications from the rebuttal.